# Graphene Oxide-Based Memristive Logic-in-Memory Circuit Enabling Normally-Off Computing

**DOI:** 10.3390/nano13040710

**Published:** 2023-02-13

**Authors:** Yeongkwon Kim, Seung-Bae Jeon, Byung Chul Jang

**Affiliations:** 1School of Electronic and Electrical Engineering, Kyungpook National University, 80 Daehakro, Bukgu, Daegu 41566, Republic of Korea; 2Department of Electronic Engineering, Hanbat National University, 125 Dongseo-daero, Yuseong-gu, Daejeon 34158, Republic of Korea; 3School of Electronics Engineering, Kyungpook National University, 80 Daehakro, Bukgu, Daegu 41566, Republic of Korea

**Keywords:** memristor, logic-in-memory circuit, in-memory computing, graphene oxide

## Abstract

Memristive logic-in-memory circuits can provide energy- and cost-efficient computing, which is essential for artificial intelligence-based applications in the coming Internet-of-things era. Although memristive logic-in-memory circuits have been previously reported, the logic architecture requiring additional components and the non-uniform switching of memristor have restricted demonstrations to simple gates. Using a nanoscale graphene oxide (GO) nanosheets-based memristor, we demonstrate the feasibility of a non-volatile logic-in-memory circuit that enables normally-off in-memory computing. The memristor based on GO film with an abundance of unusual functional groups exhibited unipolar resistive switching behavior with reliable endurance and retention characteristics, making it suitable for logic-in-memory circuit application. In a state of low resistance, temperature-dependent resistance and I-V characteristics indicated the presence of a metallic Ni filament. Using memristor-aided logic (MAGIC) architecture, we performed NOT and NOR gates experimentally. Additionally, other logic gates such as AND, NAND, and OR were successfully implemented by combining NOT and NOR universal logic gates in a crossbar array. These findings will pave the way for the development of next-generation computer systems beyond the von Neumann architecture, as well as carbon-based nanoelectronics in the future.

## 1. Introduction

Artificial intelligence (AI) and Internet-of-things (IoT) technologies have revolutionized user-friendly novel electronics, such as self-driving cars and personal cyber secretaries. However, these applications require an enormous amount of data-intensive processing which is mostly performed in the energy-hungry data center. Global carbon neutrality standards are being implemented because the massive carbon dioxide emissions from data centers contribute to global climate change. To reduce the energy consumption of the data center, the data-intensive processing should be energy-efficiently on the edge device with minimal assistance from the data center. In the modern electronic system, however, conventional von Neumann architecture suffers not only from data movement between the central processing unit and memory [1], but also normally-on computing with high static power consumption. It is important to develop an advanced computer architecture different from the von Neumann architecture.

Processing-in-memory or in-memory computing is one of the possible technologies for the advanced computing architecture. Many researchers have demonstrated the viability of in-memory computing utilizing complementary metal-oxide-semiconductor (CMOS) circuit-based technology since it was first proposed in the 1960s [2,3,4]. However, their commercialization has failed so far due to their higher cost and poor performance compared to the improved CMOS technology enabled by Moore’s law. In light of the recent abandonment of Moore’s law due to economic cost concerns [5], there is a chance for novel electronic device-based advanced computing to take its place.

The memristive logic-in-memory circuit is a cutting-edge device capable of in-memory computing [5,6,7,8,9,10,11]. Memristor [12] and memristive devices [13] were demonstrated by Hewlett-Packard (HP) in 2008 [14], as Chua’s predicted fourth basis circuit element. The unique feature of memristor is its non-volatile latching switching depending on the current or voltage history across it, enabling Boolean logic operation through in-memory computing. Particularly, the memristive logic-in-memory circuit can perform normally-off computing with the static power consumption of 0 W [15]. First, the memristive logic-in-memory circuit is proposed by the material implication (IMP) logic gate, which utilizes resistance as the logical state [6,7,11]. In the IMP logic gate, the input and output stored in the memristor array perform the basic Boolean functions by the combining of the IMP and FALSE operations, with the FALSE operation always yielding the logical ‘0’. Various logic gates and their cascading methodologies for realization of adders have been demonstrated [16,17,18]. Nonetheless, this architecture requires two sequential voltages to perform logic functions and an additional resistor within each row of the array, leading to a complex control circuit and huge power consumption. Furthermore, the output of logic function is not stored by an output memristor, but rather in one of the input memristors. To resolve the limitations of IMP logic, memristor-aided logic (MAGIC) architecture was proposed [7,10,19,20]. In the MAGIC logic architecture, the input and output memristors are separated, and the output result of logic function is stored in a dedicated memristor without an additional resistor. However, the non-uniform resistive switching of the memristor may lead to the implementation of simple gates.

Ultrathin 2D layered materials with unique benefits such as graphene [21,22,23], transition metal dichalcogenides (TMDs) [24,25,26], hexagonal boron nitride (hBN) [27,28,29], and black phosphorous [30,31], have been spotlighted due to their unique features compared to bulk materials. In particular, the graphene oxide (GO)-based memristor is of great interest due to its cost-effective solution process, controllable film thickness, superior flexibility, and transparency [32,33]. Additionally, the optimized oxygen functional group during the chemical exfoliation process can improve the switching uniformity of the GO-based memristor. Therefore, it is necessary to develop the GO-based memristor to realize ultra-low power consumption using MAGIC architecture for data-intensive computing applications.

Here, we experimentally realized the GO-based memristive logic-in-memory circuits with the static power consumption of 0 W using MAGIC logic architecture. The GO-based memristor with a structure of Ni/GO/Au showed not only uniform switching, but also stable endurance and retention characteristics. By investigating the temperature-dependent I-V characteristics and cell area-dependent switching, the switching mechanism of the GO-based memristor can be attributed to the reversible formation and rupture of metallic Ni filaments. With a GO-based memristor, we performed the basic Boolean functions such as NOT and NOR logic gates through in-memory computing. In addition, we succeeded in realizing AND, NAND, and OR logic gates in the memristor array, thus demonstrating promising potential for the memristive logic-in-memory circuit enabling normally-off computing.

## 2. Materials and Methods

GO film was synthesized from graphite using the modified Hummers method. First, graphite powder was sonicated in water to achieve exfoliated GO sheets, followed by centrifugation of the slurry of GO to remove unexfoliated particles. Ni/GO/Au devices were fabricated with crossbar structures, as shown in Figure 1a. To fabricate the Ni/GO/Au device, Au (50 nm)/Cr (5 nm) bottom electrodes (BEs) as inert and adhesion metals were patterned on SiO_2_ (300 nm)/Si substrate by the thermal evaporation process via metal shadow masks. Before depositing the GO film on BEs, UV/ozone treatment was performed. Subsequently, 90 nm-thick GO film was then deposited on the Au/Cr BEs using the spin-coating method, and it was annealed at 100 ℃ for 30 min. Next, 50 nm-thick Ni top electrodes (TEs) were formed via the thermal evaporation, perpendicular to BEs for the crossbar structure. The line widths of both TEs and BEs were identical at 60 µm. The transmission electron microscopy (TEM) pictures were captured with JEOL ARM300F equipment (Tokyo, Japan). Using a focused ion beam (FIB), a TEM sample was created for a cross-sectional analysis (FIB, FEI Helios Nano Lab 450 HP, Luxembourg, EU). The electrical characterization was performed by applying a voltage to the Ni TE, while the Au BE was grounded using a Keithley 4200 semiconductor parameter analyzer in the air environment. For the MAGIC logic operation, we utilized the built-in pulse voltage module in the Keithley 4200 semiconductor parameter analyzer. The built-in pulse voltage module, which is the pulse mode of the ramped voltage, can generate the pulse voltage with a pulse width of few ms. After the logic operation using this pulse voltage with a width of 5 ms, we measured the logical state of the output memristor and the input logical state of the input memristor using the voltage sampling mode. The logic gate operation results depending on input values were obtained sequentially according to the input values.

## 3. Results and Discussion

Figure 1a schematically shows the GO-based memristor with the crossbar array on the Si/SiO_2_ substrate. The GO-based memristor contained Ni TE, GO as the resistive switching material for the memristor, and Au TE. Raman spectra of the prepared GO film exhibited a G band at ca. 1605 cm^−1^ and a D band at 1353 cm^−1^, which correspond to the E_2g_ phonon of the sp^2^ C atoms and the disorder by certain defects [34], respectively (Figure 1b). In light of the fact that a typical GO film has a high band intensity ratio, I_d_/I_g_, larger than 1, it is noteworthy that our GO film exhibited 0.89 of I_d_/I_g_, indicating pure film characteristics. This is a result of the sophisticated centrifugation during the GO synthesis step for achieving pure GO films. In contrast to conventional GO film, the prepared pure GO film enabled the memristor device to work with uniform switching, as will be discussed in more detail subsequently. The uniform 90 nm-thick GO film between the Ni TE and Au BE was confirmed through transmission electron microscopy (TEM), as shown in Figure 1c. Figure 1d illustrates the current-voltage (I-V) characteristic of a representative GO-based memristor measured with a direct current (DC) voltage sweeping mode. Our GO-based memristor device required an electroforming process with a high voltage and compliance current to initially form the metallic filament inside the GO film, which is the phenomenon of the inherent filament-type memristor. After the electroforming process, the GO-based memristor showed a unipolar resistive switching behavior in which both SET and RESET processes occurred regardless of the voltage polarity. The positive voltage was applied to Ni TE with the compliance current, while the Au BE was grounded. When the applied voltage arrived at approximately 1.5 V, the current abruptly increased, which was designated as the SET voltage. This SET process arose from the Ni filament formation into the GO film by an electrochemical reaction. Afterward, the current rapidly decreased at around 0.5 V by re-sweeping the applied voltage to TE, which is defined as the RESET process. This RESET process occurred as a result of the joule heating-based Ni filament rupture originating from the high currents flowing through the Ni filaments. The GO-based memristor had a memory window above 10^2^ at 0.2 V read voltage, which is a low resistance state (LRS) and high resistance state (HRS) ratio. The memristor device exhibited a stable endurance feature without noticeable degradation during the repeated DC sweep mode of 300 cycles and reliable retention time over 10^4^ s without noticeable degradation (Figure 1e). Note that the reliable non-volatile characteristic of the memristor enabled normally-off computing with 0 W of static power consumption. In addition, dynamic power consumption of the memristor is one of the key performance indicators of logic gate. With the I_RESET_ of 5 mA and V_RESET_ of 0.5 V of the GO-based memristor, the dynamic power consumption can be calculated as about 2.5 mW. Considering our GO-based memristor device was not optimized in terms of power consumption, we therefore believe that the dynamic power consumption of our device could be effectively improved by using a bi-layer structure [35]. Reducing the power consumption of GO-based memristors by device structural engineering is beyond the scope of this study; therefore, further work on the topic is ongoing.

As shown in Figure 1f, the cumulative distributions of HRS/LRS and V_SET_ and V_RESET_ for the GO-based memristor showed narrow distributions. In addition, the device-to-device distribution for V_SET_ and V_RESET_ and HRS/LRS of 25 GO-based memristor devices in a crossbar array exhibited a uniform distribution without overlap (Appendix A). To minimize the sneaky current effect which causes reading errors, we performed the SET and RESET process on a specific single device by programming all devices into HRS, because our GO-based memristor 5 × 5 array had no selector device. Although this time-consuming operational method was sufficient to minimize the sneaky current in the small array size of 5 × 5, the integration of the selector device was necessary for the high size of the memristor array. These results are attributed to the pure GO film characteristics with minimal defects. Therefore, the reliable and uniform switching performance of the GO-based memristor makes it suitable for implementing a memristive logic-in-memory circuit.

To investigate the switching mechanism, the I-V curve was replotted on a double logarithmic scale, as shown in Figure 2a. The HRS region can be divided into two regions: the Ohmic region (I ∝ V) and a square-law dependence region (I ∝ V^2^). This can be elucidated by trap-controlled space charge-limited conduction (SCLS) which occurred in the ruptured filament-free region [36,37]. In contrast, the Ohmic conduction behavior was confirmed over the entire LRS region, indicating the general conductive filament model [37,38]. Moreover, the temperature dependency of the resistance in the LRS region was examined to determine the composition of the metallic filament (Figure 2b). It was confirmed that the resistance in LRS increased as the temperature rose, indicating that the conductive filament was formed by the penetration of metal ions rather than the semiconducting properties of the reduced GO film resulting from the detachment of oxygen functional groups. Note that the resistance of metal was dependent on temperature which can be fitted to $R\left(T\right)=R_{0}\left\{1+\alpha \left(T-T_{0}\right)\right\}$, where α is the temperature coefficient of resistance well-known as the intrinsic material parameter, and $R_{0}$ is the resistance at room temperature ($T_{0}$). We extracted α to be $\left(5.70\pm 0.28\right)\times 10^{-3}\;K^{-1}$, which is similar to that of Ni (about $5.86\times 10^{-3}$) [39], as shown in the inset of Figure 2b. In addition to temperature-dependent resistance characteristics, the cell area-dependent resistance feature can also support the localized conductive filament model. As shown in Figure 2c, the relatively weak dependency of resistance on the cell area in the LRS compared to that in the HRS indicated the existence of localized conductive filament in the active region. These results reveal that the GO-based memristor operated the reversible formation and rupture of the localized conductive Ni filament. The switching mechanism for the GO-based memristor can be schematically elucidated, as shown in Figure 2d. GO films decorated with various oxygen functional groups were sandwiched between the Ni TE and Au BE. When the positive bias was applied to Ni TE, the electroforming and early SET process oxidized the Ni atoms to Ni cations, and these metal ions created the localized Ni conductive filament by reduction to metallic Ni by the injected electron near the Au BE. The paths to migrate the metal ions were formed by the oxygen-deficient region which occurred by the high electroforming process. The RESET process can be explained simply by the rupture of the localized thin Ni filament due to the high local joule heating.

Figure 3a and Figure 3b illustrate the operation methodology of NOT and NOR logic gates using the MAGIC logic architecture, respectively. The MAGIC logic gates require two sequential steps. First, the output memristor is initialized to logical ‘1’ which corresponds to the resistance in LRS. In contrast, logical ‘0’ is the resistance in HRS. In the second step, the voltage V_0_ is applied to the TEs of the input memristors while the output memristor is grounded. The logical state of the initialized output memristor is dependent on a specific combination of input memristors by the voltage divider effect during the application of V_0_. Note that the voltage drop across the output memristor must be high enough for its logical state to change. When both the input and output memristors have logical ‘1’, the voltage applied to the output memristor is V_0_/2; therefore, V_0_/2 must be larger than V_RESET_. Given the cycle-to-cycle and device-to-device distributions, V_0_ is suitable for about 1.4 V. Considering the voltage variations induced by unexpected non-desired effects such as process variation and line resistance, we selected 2 V as the pulse V_0_ with a pulse width of 5 ms. However, the V_0_ of 2 V could affect the logical state of the input memristor to be changed during MAGIC implementation when the input memristor had logical ‘0’ and V_SET_ of 1 V, because the applied voltage V_0_ was applied to the input memristor. To accurately find the pulse voltage condition such as rising time, falling time, and width for MAGIC implementation, it was necessary to utilize the oscilloscope and pulse generator. Considering that the cycle-to-cycle and device-to-device distribution for V_RESET_ resulted from the ramped DC voltage, the V_0_ of 2 V was not optimal for MAGIC implementation. However, for convenience of measurement, we employed this V_0_ voltage. In addition, the ratio of HRS and LRS resistances of the memristor must be considered for reliable MAGIC logic operation. When all input memristors have logical ‘0’, the V_0_ voltage should be primarily applied to the input memristor through the voltage divider effect so as not to change the logic state of the output memristors. Therefore, the resistance ratio of HRS and LRS of the memristor must be at least 100 times higher, indicating that the GO-based memristor with an on/off ratio over 100 was suitable for reliable memristive logic operation.

Figure 3c and Figure 3d show the results of NOT and NOR MAGIC logic gates, respectively. The NOT gate was made of two memristors: one at the input, and one at the output. These two memristors were connected in series with the opposite polarity. To realize the NOT logic gate, two processes were required. After setting the output memristor to a logical “1”, voltage V_0_ was applied to the input memristor with the grounded output memristor. Input and output memristors were connected via a voltage divider generated by the applied voltage V_0_, which caused the output memristor’s state to change in response to the input memristor’s logical state. For the input memristor with logical ‘1’, the voltage drop across the output memristor exceeded its V_RESET_, resulting in a transition from logical ‘1’ to logical ‘0’ in the output memristor state. In contrast, when the input memristor was set to logical ‘0’, the applied voltage V_0_ dropped primarily across the input memristor, leading to maintaining logical ‘1’ of the output memristor state (Figure 3c).

The NOR logic gate was also implemented using the GO-based memristor, as shown in Figure 3d. To perform the NOR logic gate, parallel connections were made between the two input memristors and the output memristor. Similar to the NOT operation, the NOR operation required two execution phases. During the initial execution, the logical ‘1’ was written to the output memristor. Voltage V_0_ was applied during the second execution to induce the conditional logic operation of the output memristor. In the event that both input memristors were logical ‘0’, i.e., the ‘00’-state, the voltage drop across the output memristor was less than its V_RESET_, which maintained the initialized output memristor of logical ‘1’. In contrast, for all other input combinations, i.e., the ‘01’-, ‘10’-, and ‘11’-states, the voltage drop across the output memristor was greater than its V_RESET_, thus switching the output memristor to logical ‘0’. Given that the GO-based memirstor had enough non-volatile characteristics, the memristive logic-in-memory circuit based on the GO film can perform the normally-off computing with a static power consumption of 0 W. Therefore, the NOT and NOR gates were implemented using the data stored in the GO-based memristor on the crossbar array, enabling in-memory computing with normally-off.

Furthermore, the non-volatility of the output memristor allowed its state to be used as an input for the other logic gate, indicating its feasibility for cascade logic operations required in functionally complete logic circuits. Using the GO-based memristor, additional Boolean functions, such as OR, AND, and NAND gates, can be implemented using a network of NOT and NOR gates, known as the universal logic gates. The OR logic gate was performed through three steps, as shown in Figure 4a. A logical ‘1’ was written to the output memristor, and the input values were written to the input memristors, as part of the initialization process for the MAGIC operation. Second, the NOR logic operation with input memristors A and B was performed by applying voltage V_0_ to input memristors and ground to output memristor. The output memristor of the NOR logic gate was used as an input for the subsequent step. Lastly, the OR logic gate was successfully implemented by performing NOT logic using the output memristor of NOR. For the NOT logic operation, the voltage V_0_ was applied to the output memristor used for the previous NOR logic gate, and ground was applied to the output memristor for the NOT logic gate, resulting in realization of the OR gate. The AND gate was realized in three stages (Figure 4b). The input memristors were initialized before the MAGIC operation. Afterwards, input memristors A and B were subjected to the NOT logic operation by applying V_0_ to the input memristors. The AND logic operation was completed by connecting the two NOT gate output memristors to the NOR gate’s input memristors.

To implement the AND logic gate by performing NOR logic, voltage V_0_ was applied to two output memristors of previous NOT logic gates. As shown in Figure 4b, the AND gate was successfully implemented after the NOR operation. Figure 4c demonstrates that the output of the AND logic gate was used to implement the NAND logic gate. The output of the AND logic gate was connected to the input of the NOT gate to realize the NAND logic gate. The output memristor used for the previous AND logic gate was utilized as the input memristor for the NOT logic gate and the voltage V_0_ was applied to the input memristor. The measurement results validate the successful implementation of the NAND logic gate. Thus, we demonstrated that our GO-based memristor can realize the basic Boolean functions such as NOT, NOR, OR, AND, and NAND logic gates by cascade logic operation, enabling more efficient data processing than the other memristive logic architecture that employs external components.

It is important to compare the important device parameters of the GO-based memristor with the state-of-the art memristive logic-in-memory circuit. As demonstrated in Appendix A, the performance of GO-based memristive logic-in-memory circuits was comparable to those based on polymer, inorganic, and 2D material. Among them, initiated chemical vapor deposition (iCVD)-based polymer memristors, which were the results of our research group, showed excellent cycling endurance and retention characteristics; however, due to the inherent properties of polymer materials, it was difficult to have excellent characteristics at ultrathin thickness and high thermal stability such as in 2D materials. Given that the GO-based memristor can be easily engineered by controlling the functional groups of GO film, GO film may provide a viable platform with a high degree of freedom for developing low power and reliable memristor devices. In addition, the easy and low-cost fabrication on flexible substrate and the outstanding flexibility of GO films make the GO-based memristor suitable for flexible electronics with stylish form factors, such as smart band and curved display. Although ultrathin oxide materials with sub-10 nm thickness can be mechanically bent, these ultrathin materials-based memristors suffer from poor endurance [40]. Therefore, GO films with exceptional dislocation properties are suitable to develop low power, reliable memristive logic-in-memory circuits for the upcoming IoT era.

## 4. Conclusions

In conclusion, we demonstrated the feasibility of a GO-based memristor for realizing a memristive logic-in-memory circuit enabling normally-off computing. The fabricated GO-based memristor showed unipolar resistive switching with an on/off ratio (>10^2^), reliable endurance, retention, and uniform resistive switching due to its pure film characteristics. Based on its temperature- and area-dependent resistance characteristics, the operational principle of the GO-based memristor was the reversible formation and rupture of the Ni filament. With the GO-based memristor, we succeeded in implementing basic Boolean functions such as NOT, NOR, OR, AND, and NAND logic gates using MAGIC logic architecture. These results demonstrate that the GO-based memristor can be used to implement a much more complex, integrated logic-in-memory circuit. It is noted that the memristive logic-in-memory circuits using MAGIC architecture have two main advantages compared to conventional complementary metal oxide semiconductor (CMOS) logic circuits. First are the non-volatile characteristics of the memristor device, which enable logic computation and memory function on the same device. This logic-in-memory operation can realize the standardized logic circuit with ultralow dynamic power and a short interconnection delay compared to CMOS circuits. The extended length of the global interconnection in the advanced very large-scale integrated circuit (VLSI) is detrimental to the CMOS circuits, leading to an increase in dynamic power consumption and RC delay [41]. Moreover, the non-volatile logic-in-memory circuit can achieve the static power consumption of 0 W during standby mode, whereas the CMOS circuits composed of the volatile transistor suffer from the subthreshold leakage current. The second is a sequential cascading operation with reconfigurability. The memristive logic-in-memory circuit can implement the logic circuits with a much smaller area compared to the CMOS circuit, in which logic circuits depend on the specific gate topology to realize a specific logic function. Therefore, we believe that the GO-based memristive logic-in-memory circuit can overcome the limitations of conventional von Neumann architecture and perform a huge number of data-intensive processing for IoT and AI applications.

## Figures and Tables

**Figure 1 nanomaterials-13-00710-f001:**
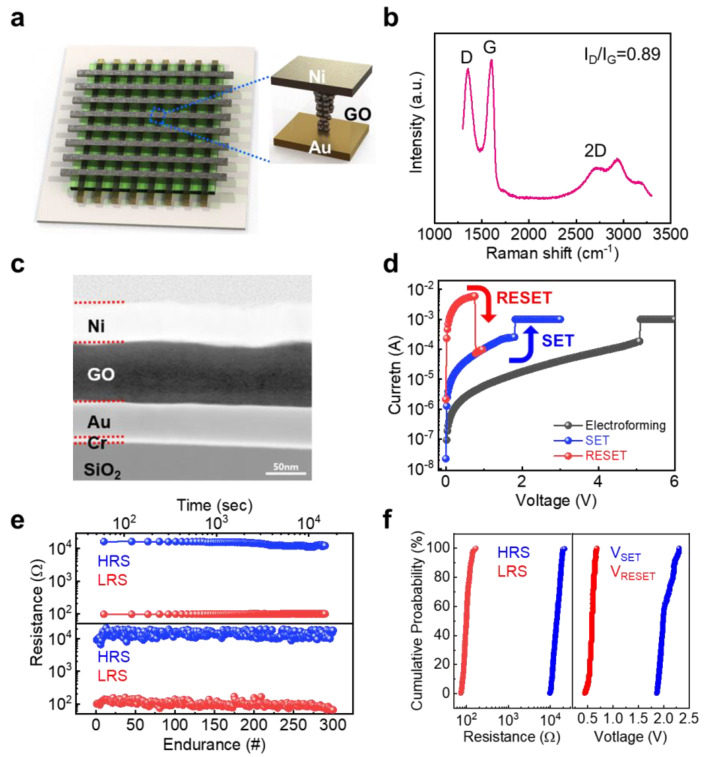
(**a**) Schematic illustration of GO-based memristor array. (**b**) Raman spectra of the prepared GO film. (**c**) Cross-sectional TEM image of GO-based memristor. (**d**) A typical I-V characteristic, (**e**) cycling endurance and retention characteristics, and (**f**) operational switching uniformity of HRS/LRS and SET/RESET voltages of GO-based memristor.

**Figure 2 nanomaterials-13-00710-f002:**
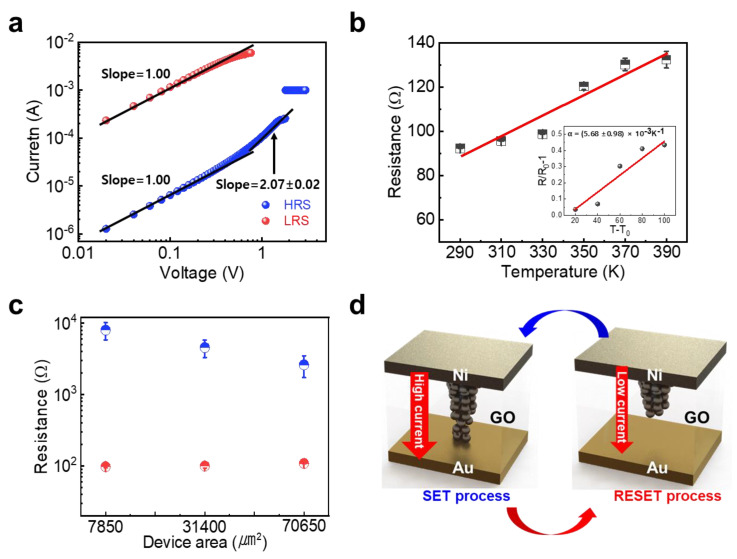
(**a**) I-V characteristics plotted on double-logarithmic scale, (**b**) temperature-dependent resistance in LRS, and (**c**) area-dependent resistance of GO-based memristor. (**d**) Schematic of the operational switching mechanism of GO-based memristor.

**Figure 3 nanomaterials-13-00710-f003:**
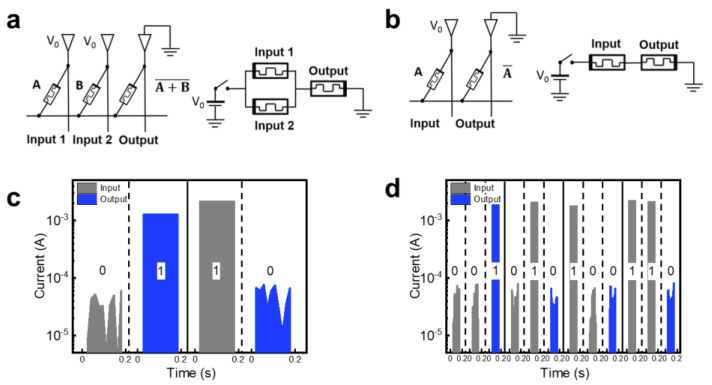
MAGIC logic gates. (**a**) NOT logic gate within memristor crossbar array and its equivalent circuit. (**b**) NOR logic gate within memristor crossbar array and its equivalent circuit. (**c**) Experimental result of NOT logic gate and (**d**) NOR logic gate of GO-based memristor. Note that the resulting values of the four NOR logic gates are the values for the combination of the four input values, and the MAGIC logic was implemented by simultaneously applying the voltage V_0_ to the two input values instead of sequentially.

**Figure 4 nanomaterials-13-00710-f004:**
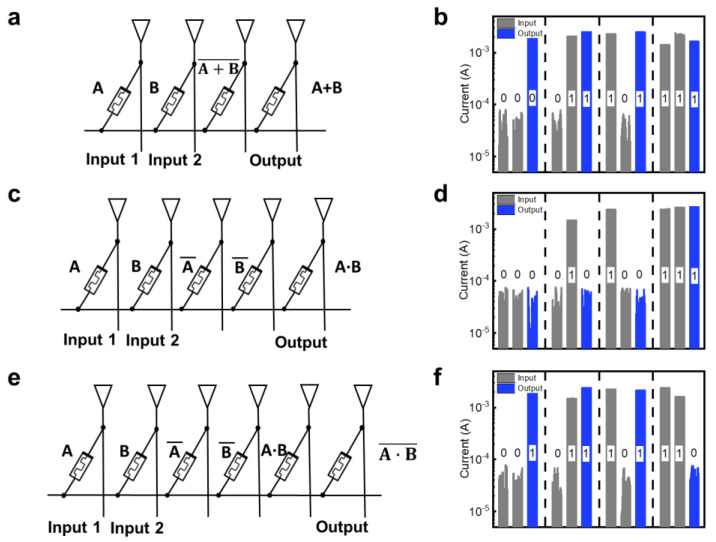
(**a**) OR logic gate within memristor crossbar array and its equivalent circuit. (**b**) Experimental result of OR logic gate. (**c**) AND logic gate within memristor crossbar array and its equivalent circuit. (**d**) Experimental result of AND logic gate. (**e**) NAND logic gate within memristor crossbar array and its equivalent circuit. (**f**) Experimental result of NAND logic gate.

## Data Availability

Not applicable.

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
