# Peer review of "Graphene Oxide-Based Memristive Logic-in-Memory Circuit Enabling Normally-Off Computing"

_nanomaterials, 2023, doi:10.3390/nano13040710_

Round 1

Reviewer 1 Report

In this paper, the authors demonstrated the GO-based memristor for realizing memristive logic-in-memory circuits, such as NOT, NOR logic gates using MAGIC logic architecture. The fabricated GO-based memristor showed unipolar resistive switching with an on/off ratio (>100), reliable endurance, retention, and uniform resistive switching.

The paper looks interesting and well-written. It would be better if the author can quantitatively compare the proposed GO-based memrister with other state-of-the-art memristors.

There are quite a few typos in the manuscript and please proofread more carefully. For example,

On line 60, “the input and output stored in the memristor array perform the basic Boolean functions by comibining of the IMP” should be "combining".

On line 70, “the non-uniform reisistive switching of memristor may lead to the implementation of simple gates” should be "resistive".

Reviewer 2 Report

This manuscript reports logic-in-memory circuit demonstration using graphene oxide based memristor devices. It covers device characteristics, analysis of conduction mechanisms, and circuit demonstration for NOT, NOR, OR, AND, and NAND. This is a comprehensive summary of this novel material set for logic-in-memory applications and the reviewer thinks that readers would be interested. However, logic-in-memory has been previously demonstrated with more conventional metal oxide based memristor and the advantage of the proposed material over prior work is not clearly explained in the current manuscript. The reviewer suggests that benchmark with prior logic-in-memory work should be included and the advantage should be clearly highlighted.

Reviewer 3 Report

Authors present a MAGIC (memristor-aided logic) implementation with graphene oxide (GO) based memristors. Although previous works have demonstrated the feasibility of this memristor logic ([1]) even with two dimensional materials ([2]), to the best of my knowledge, it is the first demonstration using GO memristors. Therefore, in my opinion, although the work shouldn´t be published in his current form, it could deserve publication in nanomaterials if the following issues were solved/clarified:

1.       Authors have shown a standard characterization of the devices (fig. 1), including variability and endurance. That is fine. Regarding the resistive switching operation, the characterization is based on a voltage ramp excitation (fig. 2). However, the device operation in the proposed MAGIC implementation is based on pulses. Therefore, I missed some characterization about the writing/erasing necessary pulses (amplitude, width) for switching the devices. Set and reset voltages can depend on the slope of ramped signals or on the pulse width in pulsed operation. In fact, the condition authors impose for V0/2=1V (“hence V0/2 must be larger than VRESET. We selected 2V as…”), which is only slightly higher than the reset voltage in Figure 1d (not to mention the effect of variability on that value), for ramped voltage operation.

2.       Regarding the previous issue, I missed an analysis about the selection of a proper value for V0 and the stability of the input memristors during operation (as Ref. [1] does). For example, let me suppose the NOR gate with A=B=0 (input memristors in the HRS). Under the application of the voltage V0=2V, the output voltage should not change, but could this voltage change the state of memristor A or B as a non-desired effect? I think that the answer could be “yes” because most of the voltage drops on the input memristors and V0 is very close (if not equal) to the set voltage. In this case, the logic gate would not work properly.

3.       I missed information about how to operate the cross-bar configuration. I mean, how to avoid sneaking paths and disturbing other memristors.

4.       From my point of view, Figs. 3c and 3d are not exact. This kind of plots can be useful as an sketch for introducing the MAGIC idea (as in reference 10 in the manuscript), but I find them little precise for a quantitative description of the final implementation. For example, please, indicate time labels for the x-axis. Why is the maximum current levels flat (without the variability that is observed in the low current levels)? And finally, in real operation, I suppose that signal Vo is applied on the input memristors simultaneously, not sequentially as it is shown in plots 3c and 3d.

5.       Consider including refs. [1]-[2] in your manuscript.

6.       There are several errata:

a.       Line 52. Maybe, “a” should not be here.

b.       Line 90, normllay

c.       Line 111: a dot is missing

d.       Figure 1f: Labels Vset and Vreset are changed, or the colour for the plots.

e.       Line 159: “s” after the equation should not be there.

f.        Line 168: “v”

g.       Figure 3: labesl “a” and “b” do not correspond to figure caption (a is the nor gate and b, the not gate).

h.       Line 221: memirstor

[1] B. Hoffer et al, Experimental Demonstration of Memristor-Aided Logic (MAGIC) Using Valence Change Memory (VCM), IEEE Transactions on Electron Devices , Vol. 67, No. 8, p. 3115-3122, 2020. DOI: 10.1109/ted.2020.3001247

[2] B. Liu et al, Bi2O2Se-Based Memristor-Aided Logic, ACS Applied Materials and Interfaces , Vol. 13, No. 13, p. 15391-15398, 2021. DOI: 10.1021/acsami.1c00177

Reviewer 4 Report

The work titled “Graphene oxide-based memristive logic-in-memory circuit enabling normally-off computing by Yeongkwon Kim et al. presented a kind of nonvolatile logic-in-memory circuit based on graphene oxide memristor which enables normally-off in-memory computing. The work is interesting. However, the following questions need to be addressed carefully before publication.

1.      There are some typos in the whole text. For example, in line 109, Au should be the BE, not the TE; In line 136, Fig. 1d can only show the I-V curve of the device during one DC operation rather than the characteristics under 300 cycles. Should it be Fig. 1e here? Figure 3a should be a NOR logic circuit, and 3b should be the NOT logic circuit, which is contrary to the note written in the article. Similarly, there are some errors in lines 193 and 205. Please check the full text carefully.

2.      How about device-to-device variation?

3.      How about the resistance requirements for the devices used in the logic circuit? For example, do devices need to have similar resistance values? Could the authors discuss more?

4.      For cascade logic operations, when  serves as the output terminal first. Does it need to be grounded or connected to V0 as shown in Fig. 4a? If the output is connected to V0, how does the primary circuit implement NOR logic?

5.      If the static power consumption of the logic circuit is 0, how about the power consumption in operation? Could the authors quantitatively evaluate this?

Round 2

Reviewer 3 Report

Although authors have improved the manuscript and answered many of the reviewers’ comments, I feel that some of the concerns I exposed in my first report have not been answered properly.

·         Regarding comment 1, I understand authors’ reasoning. But it is still based on set/reset voltages measured under ramped voltages (I assume that is the case for the CDFs shown in figures 1f, S1a and S1b), while the operation of the gates is performed by means of pulses. Set and reset voltage could be different for ramped or pulsed operation regimes (even for voltage ramps with different slopes).

·         Regarding comment 2, the first part of the authors’ answer is based on a V0 value (1.4 V that they do not use in the practical operation of the gates) and they agree with this reviewer about the fact that the actual used value of Vo (2V) could affect the input memristors. Please, indicate if that fact has been observed and how it could affect to a final implementation of the gates. Please, add that information in the manuscript.

·         Regarding my question about sneaking paths (comment 3), I agree with the authors´ answer, but I think that that information has to be also translated to the manuscript.

·         Regarding comment 4, the authors’ answer is confusing. Authors claim in their answer: “the V0 pulse voltage with width of 5 ms is only applied to the input memristor simultaneously”. However, in figure 3d it still appears that the application of the voltage pulses on the input memristors is sequential. If figure 3d corresponds to the determination of the input memristors after the logic operation of the gate (as the text added by authors seems to mean), please indicate that in the figure 3c and 3d captions.

Furthermore, the labels in the x-axis have not been added (I asked for that in my previous review). The time unit is “sec”, but there are no tick labels.

Reviewer 4 Report

The authors have processed my previous comments. I have no more questions.

Round 3

Reviewer 3 Report

Authors have answered my questions and concerns. From my point of view, the paper can be published in its current form.